# Subgoal-Based Explanations for Unreliable Intelligent Decision Support Systems

**Devleena Das** [1]**, Been Kim,** [2]**, Sonia Chernova,** [1]

[1]Georgia Institute of Technology, Atlanta, GA, USA
[2]Google Research, Mountain View, CA, USA
ddas41@gatech.edu, beenkim@google.com, chernova@gatech.edu

## Abstract

Intelligent decision support (IDS) systems leverage artificial intelligence techniques to generate recommendations that guide human users through the decision making phases of a task. However, a key challenge is that IDS systems are not perfect, and in complex real-world scenarios may produce suboptimal output or fail to work altogether. The field of explainable AI planning (XAIP) has sought to develop techniques that make the decision making of sequential decision making AI systems more explainable to end-users. Critically, prior work in applying XAIP techniques to IDS systems has assumed that the plan being proposed by the planner is always optimal, and therefore the action or plan being recommended as decision support to the user is always optimal. In this work, we examine novice user interactions with a non-robust IDS system – one that occasionally recommends a suboptimal actions, and one that may become unavailable after users have become accustomed to its guidance. We introduce a new explanation type, *subgoal-based explanations*, for plan-based IDS systems, that supplements traditional IDS output with information about the subgoal toward which the recommended action would contribute. We demonstrate that subgoal-based explanations lead to improved user task performance, improve user ability to distinguish optimal and suboptimal IDS recommendations, are preferred by users, and enable more robust user performance in the case of IDS failure.

## Introduction

Intelligent decision support (IDS) systems leverage artificial intelligence techniques to generate recommendations that guide human users through the decision making phases of a task (Sutton et al. 2020). While much prior work has focused on decision support for domain experts (e.g., cancer diagnosis for oncologists (Walsh et al. 2019)), increasingly, IDS systems have been proven particularly useful in helping *novice users* make decisions (Gutiérrez et al. 2019; Machado, Lam, and Chen 2018). However, a key challenge is that IDS systems are not perfect, and in complex real-world scenarios the actions recommended by IDS systems may be far from optimal (Guerlain, Brown, and Mastrangelo 2000). Such errors particularly strongly affect novice users, who lack the knowledge to assess the correctness of an IDS recommendation (Nourani, King, and Ragan 2020).

The field of *explainable AI Planning (XAIP)* have developed techniques to make sequential decision making systems more understandable to domain-experts. Prior work on XAIP has largely focused on explaining plan solutions that help users answer questions "Why plan P?" and "Why not plan Q?", through popular methods including causal-link-chain (CLC) explanations (Seegebarth et al. 2012) as well as contrastive explanations (Hoffmann and Magazzeni 2019). These techniques have been effective in helping domain-experts understand how their proposed solution differs from a planner's solution (Hoffmann and Magazzeni 2019), as well as how performing a current action effects the preconditions of future actions (Seegebarth et al. 2012).

Critically, prior work in applying XAIP techniques to IDS systems has assumed that the plan being proposed by the planner is always optimal, and therefore the action or plan being recommended to the user is always optimal (Grover et al. 2020; Valmeekam et al. 2020). However, optimal IDS decision making cannot be guaranteed in complex real world deployments. In fact, in real world systems, other assumed, robust characters of IDS systems may not hold true, including the ability to always receive suggestions at deployment. There may be situations in which a user's query is unanswerable, or the IDS system runs into a failure and is no longer available to the user.

In this work, we examine novice user interactions with a non-robust IDS system – one that occasionally recommends the wrong action, and one that may become unavailable after users have become accustomed to its guidance. A user of such a system, given an IDS action recommendation, must be able to determine whether the recommendation is optimal or not. In the absence of an IDS recommendation, the ideal user will have sufficient understanding of the task such that their task performance is not negatively impacted by the sudden absence of previously available recommendations. Leveraging insights from Psychology, which demonstrate that humans naturally break down large complex tasks into a smaller set of more manageable subgoals (Newell, Simon et al. 1972), we introduce a new explanation type – *subgoal-based explanations* – that supplements traditional IDS output with information about the subgoal toward which the recommended action would contribute. We then examine the impact such an explanation has on novice user performance through experiments in a com-

plex planning domain–restaurant planning–with 105 study participants. We compare our subgoal-based explanations with the traditional action recommendation outputs of IDS systems, as well as causal-link chain (CLC) explanations (Seegebarth et al. 2012), an XAIP technique most relevant to our work. We contribute several key findings:

- In the context of a suboptimal IDS system, subgoal-based explanations enable users to successfully detect and avoid more suboptimal IDS recommendations than users who are only provided traditional IDS action recommendations or CLC explanations.

- Users who receive subgoal-based explanations achieve better performance when performing a task under IDS guidance than users who receive IDS guidance with CLC or those who receive traditional IDS action recommendations.

- Users who are exposed to subgoal-based explanations for some period of time, are able to perform the task more reliably in the absence of further IDS, compared to users who only receive action recommendations or CLC explanations. This finding suggests that explanations contribute a significant training benefit beyond both traditional IDS output as well as CLC explanations.

- In a direct comparison, users exhibit a strong preference for IDS output that includes subgoal-based explanations versus CLC explanations or IDS output that includes the next action.

We also show a simple way to generate domain-independent subgoal-based explanations that can generalize to any hierarchical plan-based system and broadly applicable across a wide range of IDS systems and application areas. Together with our findings, our work is a first step towards investigating how and when IDS with XAIP systems are beneficial in complex real-world IDS systems that are not fail-proof.

## Related Work

In this section, we situate our work in the context of the two prominent research areas most closely related to our work: Intelligent Decision Support Systems and Explainable AI.

### Intelligent Decision Support (IDS) Systems

IDS systems have been developed to assist users in decision making across a wide range of applications, such as providing assistance to domain-experts in clinical settings (Walsh et al. 2019; Zhuang et al. 2009) and aiding novice-users in management settings (Machado, Lam, and Chen 2018; Gutiérrez et al. 2019). Amongst these IDS systems, decision support is provided through a range of mediums, depending on the domain. For instance, in (Machado, Lam, and Chen 2018), the authors develop a mobile app for clinical decision support that allows dental students to answer a series of questions to determine a diagnosis and provide treatment suggestions. In (Gutiérrez et al. 2019), researchers investigate how to best portray visual representations and interaction techniques to aid novice users in business decisions.

Given that many IDS systems interact with end-users who are not AI-experts, several bodies of work have investigated how to enhance the transparency of IDS systems to improve user trust. These transparency techniques have been studied in the context of explaining recommendations for single-classification tasks, such as clinical decision support to identify failure modes (Jones, Mateer, and Harrison 2019; Feng, Shaib, and Rudzicz 2020). By contrast, our work investigates IDS in sequential decision making settings. Specifically, we investigate how to provide explanations that help novice users improve their decision making performance in the presence of potentially suboptimal suggestions.

### Explainable AI

The field of explainable AI aims to improve a user's understanding of the inner workings of complex models (Doshi-Velez and Kim 2017). Given that AI and ML models are not guaranteed to be optimal, an important objective of XAI techniques includes being able to help users identify vulnerabilities or "bugs" within a model (Adebayo et al. 2020) as well as identify any spurious correlations (Kim et al. 2018). Our work explores a scenario in which the underlying AI model, and therefore the explanation that results from it, may be suboptimal. We examine how, even under these settings, users can leverage explanations to ultimately improve their task performance.

To provide greater context for our work, we first review the types of explanations that have been developed in sequential decision making. The subfield of explainable AI Planning (XAIP) seeks to develop methods for explaining sequential decision making problems, where an agent engages in a longer-term interaction with a user (Chakraborti, Sreedharan, and Kambhampati 2020). Within the community, techniques have primarily focused on explaining an agent's entire plan solutions to end-users. A recent survey by Chakraborti et al. (Chakraborti, Sreedharan, and Kambhampati 2020) highlights the key areas of plan explanations that have been investigated, including generating contrastive explanations (Hoffmann and Magazzeni 2019), explaining unsolvable plans (Sreedharan et al. 2019), and generating explicable plan explanations (Chakraborti et al. 2017).

Additionally, XAIP techniques have also been applied to plan-based decision support systems in efforts to improve human-in-the-loop planning. For example, RADAR by Grover et al. provides XAIP features such as plan summarization, plan explanations in the form of minimally complete contrastive explanations, plan validation, and action and plan suggestions to improve decision making (Grover et al. 2020). Furthermore, Valmeekam et al develop RADAR-X which leverages user queries understand user preferences for providing refined plan suggestions (Valmeekam et al. 2020). Our work similarly aims to support human-in-the-loop planning, with an important difference that we do not make the assumption of an optimal AI planner or optimal IDS systems recommendations.

## Problem Formulation

In this section, we first provide definitions for the planning problem that underlies our IDS system. We then formalize the problem of providing explanations in support of plan-based IDS and present our research hypotheses.

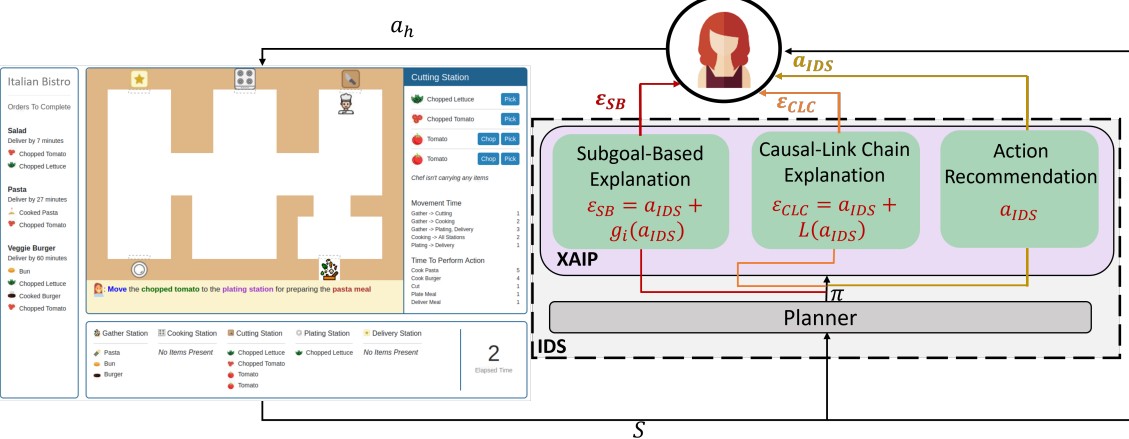

Figure 1: Our IDS system utilized to evaluate the efficacy of $\mathcal{E}_{\mathcal{SB}}$. A user performs any action $a_h$, which updates the environment $S$. The underlying planner utilizes $S$ to produce a plan solution $\pi$. The IDS system provides either $\mathcal{E}_{\mathcal{SB}}$, $\mathcal{E}_{\mathcal{CLC}}$ or $a_{IDS}$. If the user ignores the planner's suggestion, the planner will replan for a new $\pi'$, repeating the process.

## Planning Problem

A planning problem is defined by a model $\mathcal{M} = \langle \mathcal{D}, \mathcal{I}, \mathcal{G} \rangle$ where domain $\mathcal{D}$ is represented by $\langle F, A \rangle$, such that $F$ is a finite set of fluents that define a state $s \subseteq F$, and $A$ represents a finite set of actions. $\mathcal{I}$ and $\mathcal{G}$ represent the initial and goal states, respectively, such that $\mathcal{I}, \mathcal{G} \subseteq F$. Note that $\mathcal{G}$ may be modeled as a set of $\langle g_0...g_j \rangle$, where $g_i$ represents a subgoal. An action $a \in A$ is defined by a tuple $\langle c_a, pre(a), eff^+(a), eff^-(a) \rangle$, where $c_a$ is the associated cost of $a$, and $pre(a), eff^+(a), eff^-(a)$ denote the set of preconditions, add and delete effects, respectively. An action $a \in A$ can only be executed in a state $s$ if $s \models pre(a)$. A transition function, $\delta_M(s, a)$ is used to transition an agent from $\mathcal{I}$ to $\mathcal{G}$, performing a sequence of actions $\langle a_1...a_n \rangle$, each with an associated cost $c_a$. In other words, the cost of plan $C(\pi, \mathcal{M})$ is defined by $\sum_{a \in \pi} c_a$, the sum cost of all actions within the plan, or $\infty$ if the goal is not met. The solution to a planning problem is a plan $\pi = \langle a_1...a_n \rangle$ such that $\delta_M(\mathcal{I}, \pi \models \mathcal{G})$. The optimal plan solution, $\pi^*$, is defined by $argmin_\pi \{C(\pi, \mathcal{M}) \forall \pi$ such that $\delta_{\mathcal{M}}(\mathcal{I}, \pi) \models \mathcal{G}\}$.

## Explainability in Plan-Based IDS Systems

The goal of a plan-based IDS system is to provide the user with action recommendations $a_{IDS} \in \pi$. In turn, the user, who is given $a_{IDS}$ as input, must select their own action $a_h$ to take in response. In the ideal case, the IDS guides the user along some optimal plan $\pi^*$ by always recommending an optimal action, $a_{IDS} = a^* \in \pi^*$, which results in the user always taking the optimal action, $a_h = a^* \in \pi^*$. However, there are two limitations to this idealized formulation.

First, in complex real-world scenarios an IDS systems may not be able to generate an optimal plan, resulting in suboptimal recommendations (Guerlain, Brown, and Mastrangelo 2000). In this case, the user is faced with the challenge to discern whether the IDS system's recommended action is optimal (i.e., $a_{IDS} \overset{?}{=} a^*$). Relating to this, we state the following hypothesis:

**H1:** We hypothesize that there exists a type of explanation, $\mathcal{E}$, that when presented in conjunction with $a_{IDS}$ can aid users in determining $a_{IDS} \overset{?}{=} a^*$.

Specifically, that with the aid of $\mathcal{E}$, users will be able to accept $a_{IDS}$ with greater accuracy when $a_{IDS} = a^*$ and correctly reject $a_{IDS}$ when $a_{IDS} \neq a^*$.

Second, in complex real-world scenarios an IDS system may not always be available due to being offline, a failure, or the query being outside its scope. In this case, the user may suddenly be required to select $a_h$ without the benefit of an IDS system's guidance. This scenario will pose a particularly significant challenge to users who had previously only performed the task under the support of an IDS system. Relating to this, we state the following hypothesis:

**H2:** We hypothesize that exposure to explanations $\mathcal{E}$ improves user understanding of the task, such that when IDS recommendations are turned off, users with previous exposure to $\mathcal{E}$ will achieve greater task performance than users who had the same amount of domain experience but without exposure to $\mathcal{E}$.

Specifically, we posit that users previously exposed to explanation $\mathcal{E}$ will be able to select actions $a_h$ that lead to more optimal task performance ($C(\pi_h^{\mathcal{E}}, \mathcal{M}) < C(\pi_h^{\cancel{\mathcal{E}}}, \mathcal{M})$) than users who were not exposed to explanation $\mathcal{E}$. In this perspective, explanations can be seen as a *training mechanism* that leverages IDS to improve user understanding of the task.

Finally, prior work across many XAI applications has demonstrated that incorporating explanations into the output of automated systems improves user performance in a given task (Das and Chernova 2020; Tabrez, Agrawal, and Hayes 2019). Relating to this, we state the following two hypotheses in the context of IDS systems:

**H3:** We hypothesize that user performance on the task will improve when IDS output, $a_{IDS}$, is supplemented with explanation $\mathcal{E}$.

**H4:** We hypothesize that users will prefer the output

of a system that includes $\mathcal{E}$ over a system that only includes $a_{IDS}$.

Specifically, we posit that overall plan cost for users exposed to explanations will be lower than for users who did not receive explanations (i.e., $C(\pi_h^{\mathcal{E}}, \mathcal{M}) < C(\pi_h^{\not{\mathcal{E}}}, \mathcal{M})$), and that users will prefer to see explanations as part of an IDS output.

In the next sections, we first present a new variant of $\mathcal{E}$ for explaining IDS systems action recommendations, leveraging findings in psychology (Newell, Simon et al. 1972). We then describe our validation domain, and the experiments that were conducted to support the above hypotheses.

## Subgoal-Based Explanations

Research in psychology shows that humans faced with a complex sequential decision making task naturally construct a mental model of the task as a decomposition of multiple subgoals (Newell, Simon et al. 1972). Similarly, many AI techniques utilize hierarchical structures to leverage the improved computational efficiency of such representations (Iovino et al. 2020; Kaelbling and Lozano-Pérez 2010).

In this work, inspired by the natural hierarchical representations used by both human users and AI systems, we introduce a new type of explanation known as subgoal-based explanations, $\mathcal{E}_{\mathcal{SB}}$. The objective of $\mathcal{E}_{\mathcal{SB}}$ is to improve user task performance both in optimal and suboptimal IDS settings. Below we further detail the definition of $\mathcal{E}_{\mathcal{SB}}$ by leveraging the definition of a planning problem (see previous section):

> $\mathcal{E}_{SB}$: Given planning goal $\mathcal{G}$, which is decomposed into a set of subgoals $\langle g_0...g_n \rangle$, a subgoal-based explanation is described by $\mathcal{E}_{SB} = a_{IDS} + g_i(a_{IDS})$.

In other words, a subgoal-based explanation provides the next recommended action from an IDS system, $a_{IDS}$ along with the associated subgoal $g_i$ that is satisfied by $a_{IDS}$. An example of a $\mathcal{E}_{SB}$ is "Chop the tomato to prepare the lasagna," where "to prepare the lasagna" is the subgoal. Plan subgoals for explanation generation may be predefined in the planning representation, or may be autonomously derived using established methods (Richter, Helmert, and Westphal 2008; Czechowski et al. 2021). In this work, we encode the subgoals within our plan problem definition.

## IDS in Restaurant Planning Domain

We investigate the efficacy of $\mathcal{E}_{\mathcal{SB}}$ explanations in the context of a complex sequential planning scenario: running a restaurant kitchen. In our task a user plays as a chef to deliver a set of $M$ meals, each within unique delivery times, with the help of an anthropomorphized IDS known as Manager Molly. Below we further detail the restaurant planning game and our plan-based IDS system.

### Restaurant Game Overview

Figure 1 provides a visualization of the restaurant game; inspired by the online game Overcooked, this domain has been studied extensively in the sequential decision making community (Carroll et al. 2019; Wu et al. 2021; Liu et al. 2020). Within the game, the user controls a chef avatar, and utilizes five meal prep stations to prepare $M$ meals consisting of various ingredients. The game objective is to deliver the meals to restaurant customers within the designated meal prep time for each meal, denoted as $t_{goal\_delivery}^m$. The five meal prep stations are: *gather station*, *cutting station*, *cooking station*, *plating station*, and *delivery station*. The game state is represented as $S$, and displayed to the user in a user-friendly manner in the bottom panel of the game interface (see Fig. 1). Specifically, $S = \{S_l, S_i\}$ where $S_l$ defines the location of the chef and each ingredient (i.e which station), and $S_i$ defines the state of each ingredient,(i.e. chopped tomato, cooked chicken). The user is able to perform an action $a^h$ from the action space $A = \{$*cut, move-chef, move-item, start-cook, end-cook, deliver, prepare-meal*$\}$, so long as the preconditions of $a_h$ are met and that the effects of $a_h$ result in a valid state $S$. The game does not allow players to perform an invalid action (e.g., preparing the steak meal without cooking the steak). When $a_h$ is performed, game state $S$ is updated and utilized by the underlying planner to provide a recommendation to the user, either in the form of a subgoal-explanation $\mathcal{E}_{\mathcal{SB}}$, a causal-link-chain explanation $\mathcal{E}_{\mathcal{CLC}}$ (Seegebarth et al. 2012), or as a default action recommendation $a_{IDS}$. Given a recommendation, the user can choice to conform to $a_{IDS}$ or select an alternative action (i.e. $a_h \neq a_{IDS}$). If the user selects an alternative action, the underlying planner in the game utilizes the updated state information to find a new plan solution $\pi'$ from which subsequent actions are suggested to the user.

### Restaurant Game Planner

To find a plan solution $\pi$, we utilize the Temporal Fast Downward (TFD) planner (Eyerich, Mattmüller, and Röger 2009). Although TFD is a temporal planner with abilities to handle durative actions, we formulate our planning problem as a classical planning problem in which the objective is to minimize action costs as opposed to duration. We utilize TFD to more accurately model a restaurant domain where actions can occur simultaneously. To simulate multitasking, we leverage TFD's support for numeric fluents which allows us to track the cost of actions $a \in A$ being performed while an ingredient is cooking to ensure that an appropriate duration has elapsed for an ingredient to cook.

To solve for a plan solution $\pi$, the planner utilizes the same action space $A$ and state space $S$ as that available to the user. The planner's initial state $\mathcal{I}$ is defined with select, pre-performed actions to avoid plan solutions longer than 35 actions and the goal state $\mathcal{G}$ is defined by reaching the *delivered* state for all necessary $M$ meals. Each action $a \in A$ has a static cost of $c_a$ where the cost represents the time needed to perform $a$. If each meal $m$ is delivered at $t_{delivered}^m$, the objective of the planner is to minimize the overtime delivery cost, $\sum_{m=1}^{|M|} t_{delivered}^m - t_{goal\_delivery}^m$.

### Generating Suboptimal Plans

A central objective of our work is to examine explanatory action suggestions in the context of *suboptimal* IDS recommendations. To achieve this objective, we intentionally

| Action ($a \in \pi$) | Action Recommendation ($a_{IDS}$) | Causal-Link-Chain Explanation ($\mathcal{E}_{\mathcal{CLC}}$) | Subgoal-Based Explanation ($\mathcal{E}_{\mathcal{SB}}$) |
|---|---|---|---|
| cut chef gatherStation cutStation tomato1 | Chop the tomato. | Chop the tomato for the salad meal or pasta meal or veggie burger meal. | Chop the tomato for the salad meal. |
| move-item chef gatherStation plateStation salmon1 | Move the salmon to the cooking station. | Move the salmon to the cooking station to cook the salmon. | Move the salmon to the cooking station for preparing the teriyaki salmon meal. |
| end-cook chef cookStation broth1 | Finish cooking the broth. | Finish cooking the broth for preparing the soup. | Finish cooking the broth for preparing the soup. |
| move-chef chef plateStation gatherStation | Move to the gather station. | Move to the gather station to move the tomato. | Move to the gather station for preparing the pasta meal. |

Table 1: Action recommendations ($a_{IDS}$), causal-link chain explanations $\mathcal{E}_{\mathcal{CLC}}$ and subgoal explanations ($\mathcal{E}_{\mathcal{SB}}$) for select $a \in \pi$.

corrupt the optimal plan, $\pi^*$, generated by our planner such that the resulting plan $\tilde{\pi}$ is suboptimal. At run time, we randomly select with probability $p$ whether a recommended action, $a_{IDS}$, is provided from an optimal or suboptimal plan. Specifically:

$$a_{IDS} = \begin{cases} a_{IDS} \in \tilde{\pi}, & \text{if } rand() \leq p \\ a_{IDS} \in \pi^*, & \text{otherwise} \end{cases}$$

Recall, the goal of the planner in our restaurant planning domain is to minimize the overtime in delivering each meal $m \in M$. Therefore, to generate a suboptimal plan $\tilde{\pi}$, we replace the optimal action of interest required for a particular meal $m_i$ with a random action required for some other random future meal $m_j$. The resulting suboptimal action is therefore still relevant to the overall cooking task, and is not an obvious and trivially identifiable error (e.g., throw steak on floor). Reordering actions in this way is guaranteed to result in a suboptimal plan because it delays meals and leads meals to be completed out of order, causing the planner's overtime delivery cost to be non-zero.

### Generating Subgoal-Based Explanations

Given a set of subgoals $\langle g_1...g_n \rangle$, we employ a post-hoc search to map actions $\langle a_0...a_n \rangle$ within $\pi$ with a corresponding subgoal $g_i \in G$. In our work, subgoals are defined as the designated meal for which an action $a$ is being performed. To present $\mathcal{E}_{\mathcal{SB}}$ in a manner understandable by novice users, we leverage natural language. We parse each $a$ output in $\pi$ for the contextual information the action $a$ is acting upon. In our work, the contextual information corresponds to the *ingredient(s)* the action would be applied on or the *location* the action would be applied to. In this manner, we template our explanation as follows, "$\langle action \rangle$ the $\langle ingredients /location \rangle$ for $\langle g_i(a) \rangle$". Table 1 provides example explanations of our $\mathcal{E}_{\mathcal{SB}}$ explanations in comparison to action suggestions, $a_{IDS}$, and causal link chain explanations $\mathcal{E}_{\mathcal{CLC}}$. A causal link $l$ is defined as $(s \rightarrow_p s')$ which denotes that preconditions $p$ of a future plan step $s'$ are met by the effects of the current plan step $s'$ (Seegebarth et al. 2012). Thus $\mathcal{E}_{\mathcal{CLC}}$ justify the action $a$ in the current plan step by indicating the set of satisfied causal links $l \in L$. Alternatively, $a_{IDS}$ models the default output of current IDS systems.

### Study Design

Our primary goal is to evaluate the effect $\mathcal{E}_{\mathcal{SB}}$ explanations have on user task performance in IDS settings. We conducted a seven-way, between-subjects study in which participants were asked to play the restaurant planning game. Specifically, we used a 3 x 2 factorial study design with one factor being the type of IDS (with $a_{IDS}$ suggestions, $\mathcal{E}_{\mathcal{CLC}}$ or $\mathcal{E}_{\mathcal{SB}}$ explanations) and the second factor being optimality of IDS recommendation (optimal IDS and suboptimal IDS). The seventh study condition was an additional baseline condition in which participants did not receive any help from an IDS. Below, we detail each study condition:

- **None (Baseline):** Participants receive no suggestions from an IDS system.
- **$\pi(a_{IDS})$ (Baseline):** Participants receive action recommendations from an optimal IDS system, and is closely modeled after the default action suggestion features available in plan-based IDS systems.
- **$\pi(\mathcal{E}_{\mathcal{CLC}})$ (Baseline):** Participants receive causal link chain explanations from an optimal IDS system and is closely modelled after the causal link explanations in (Seegebarth et al. 2012).
- **$\pi(\mathcal{E}_{\mathcal{SB}})$:** Participants receive subgoal-based explanations from an optimal IDS system.
- **$\tilde{\pi}(a_{IDS})$:** Participants receive action recommendations from a suboptimal IDS system.
- **$\tilde{\pi}(\mathcal{E}_{\mathcal{CLC}})$:** Participants receive causal link explanations from a suboptimal IDS system.
- **$\tilde{\pi}(\mathcal{E}_{\mathcal{SB}})$:** Participants receive subgoal-based explanations from a suboptimal IDS system.

The study consisted of three stages in which participants played a total of five games, each consisting of a unique set of $M$ meals to prepare. Participants proceeded to the next game when they finished delivering all required $M$ meals, or when time cost in a game reached 80, whichever came first. The study consisted of three stages: the familiarization stage, IDS stage and an assessment stage, detailed below.

*Familiarization:* The participants first played through an interactive tutorial which explained the components of the

interface as the participants made a burrito meal. While the interactive tutorial remained the same across all conditions, the None condition did not receive support from the IDS system, whereas all other conditions received their respective guidance from the IDS system. Participants also played a second game to get further acquainted with the system. In the practice round, participants were tasked with making two meals (BLT sandwich, hotdog), and participants received IDS based on their study conditions. The familiarization stage was designed to familiarize users with all aspects of the interface and to minimize learning effects in future games.

_IDS:_ Participants played two more games, each with the objective of preparing three meals with the help from an IDS system (or no help in the None condition). These games were themed by cuisine: Italian Bistro (salad, pasta, veggie burger) and Asian Fusion (chicken quesadilla, soup, sushi). Prior to playing, participants were told that the anthropomorphized IDS system, Manager Molly, may provide suboptimal suggestions. The goals of the participants in all conditions were to delivery meals on time and to identify suboptimal suggestions (in the four conditions with IDS). Both games were counterbalanced, such that a random 50% of participants played the Italian Bistro game first, while remaining played the Asian Fusion game first. In the two suboptimal recommendation study conditions ($\tilde{\pi}(a_{IDS})$ and $\tilde{\pi}(\mathcal{E}_{\mathcal{SB}})$), $p = 0.85$, such that the accuracy of our IDS was 85%, and approximately 15% of recommendations viewed by the users were corrupted to be suboptimal [1].

_Assessment:_ Participants in all five conditions played a final game, with **no support from the IDS system**. Similar to the IDS stage, the assessment game required delivering 3 meals (teriyaki salmon, steak & potatoes, chili). In this assessment, participants in all study conditions had one objective, which was to deliver meals by their designated delivery time. The goal of the assessment stage was to simulate a scenario where a failure occurs in an IDS system, and guidance is no longer available to a novice user. Our goal is to understand how previous exposure to IDS in an optimal or suboptimal setting may impact participant performance on a task when in the absence of any IDS.

### Metrics & Hypotheses

To measure user task performance and overall understanding of the task, we evaluate three metrics:

- **User Plan Cost** (*UPC*): represents participant overtime cost in delivering meals *per game*. This metric is analogous to how the planner optimizes for the optimal plan solution. Below, $M$ represents the total number of meals to complete for a game, $t^m_{delivered}$ represents the accumulated time cost at which meal $m$ is delivered,

---

[1] The level of acceptable error in a deployable IDS system varies significantly by application (e.g., medical diagnosis systems may be expected to perform with greater accuracy than those in lower risk domains). From prior work, we find accuracy rates of 75-95% across applications (Rodríguez, Gonzalez-Cava, and Pérez 2020; Rathore, Loia, and Park 2018). We selected an accuracy of 85% as it models many state of the art systems.

and $t^m_{goal\_delivered}$ represents the designated time cost at which a meal should have been delivered.

$$UPC = \sum_{m=1}^{|M|} t^m_{delivered} - t^m_{goal\_delivery} \qquad (1)$$

- **Optimal Action Conformance** (OAC%): represents the total percentage of optimal actions suggested from the IDS system that participants performed.

- **Suboptimal Action Avoidance** (SAA%): represents the total percentage of suboptimal actions suggested from the IDS system that participants avoided.

- **Perceived Preference** (*Pref%*): represents the total percentages of $a_{IDS}$ or $\mathcal{E}_{\mathcal{SB}}$ IDS types preferred by participants for understanding the chef's next action.

Both *OAC* and *SAA* are measured during the *IDS Stage* of the user study, while *UPC* is measured for games within the *IDS Stage* and *Assessment Stage*.

### Participants

We recruited 120 individuals from Amazon's Mechanical Turk. We filtered participants that showed no effort to play the game in the form of taking repeated actions until the timeout. The filtration process yielded 105 remaining participants (53 males, 52 females). All participants were 18 years or older (M = 36.3, SD =10.3). Each study condition had 15 participants. The task took on average 40 minutes and participants were compensated $5.00.

## Study Results

The User Plan Cost (*UPC*) metric in Figure 3 and Figure 4 as well as the the Optimal Action Conformance (*OAC*) and Suboptimal Action Avoidance (*SAA*) metrics were analyzed with a one-way ANOVA, followed by a post-hoc Tukey Test. **H1:** In Figure 2, we present user optimal action conformance (*OAC%*) as well as suboptimal action avoidance (*SAA%*) in order to analyze the benefit of providing subgoal-based explanations, $\mathcal{E}_{\mathcal{SB}}$, for understanding suboptimality in IDS systems. We observe in Figure 2(a) that $\tilde{\pi}(a_{IDS})$, $\tilde{\pi}(\mathcal{E}_{\mathcal{CLC}})$ and $\tilde{\pi}(\mathcal{E}_{\mathcal{SB}})$ conditions had similarly high *OAC%*s. In other words, action recommendations, causal link chain explanations and subgoal-based explanations helped participants discern optimal recommendations. However, in Figure 2(b), we observe that participants in the $\tilde{\pi}(a_{IDS})$ condition and $\tilde{\pi}(\mathcal{E}_{\mathcal{CLC}})$ condition had much lower *SAA%*s than participants in the $\tilde{\pi}(\mathcal{E}_{\mathcal{SB}})$ condition. In fact, in the Asian Fusion game, we observe a significant difference in *SAA%* between the $\tilde{\pi}(a_{IDS})$ and $\tilde{\pi}(\mathcal{E}_{\mathcal{SB}})$ conditions (t(42)=-2.80, $p < 0.05$) as well as between the $\tilde{\pi}(\mathcal{E}_{\mathcal{CLC}})$ and $\tilde{\pi}(\mathcal{E}_{\mathcal{SB}})$ conditions (t(42)=-2.95, $p < 0.05$). **These results support H1**, indicating that in the context of an suboptimal IDS system, subgoal-based explanations, $\mathcal{E}_{\mathcal{SB}}$, help participants detect and avoid more suboptimal suggestions compared to those who only receive $a_{IDS}$ or $\mathcal{E}_{\mathcal{CLC}}$.

**H3:** In Figure 3, we present user plan costs (*UPC*) across each study condition in the two themed games to analyze the impact of including subgoal information on IDS-supported

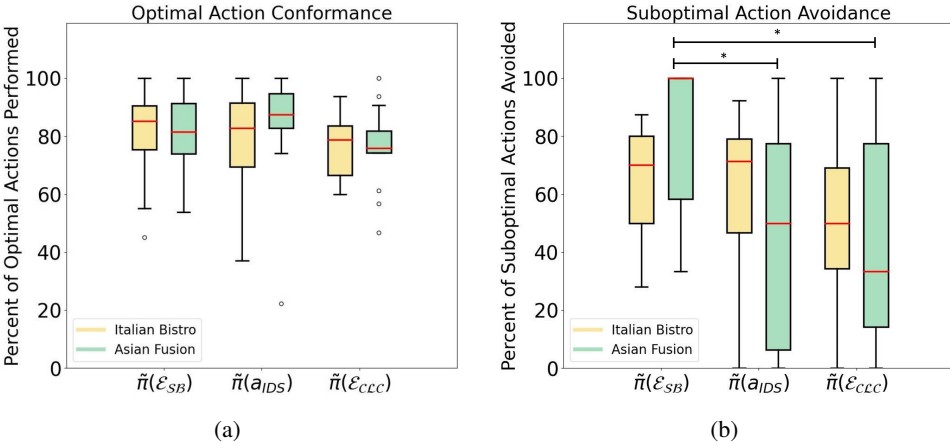

(a)

(b)

Figure 2: User optimal action conformance and action avoidance percentages for participants that received $\mathcal{E}_{\mathcal{SB}}$ and $a_{IDS}$ from suboptimal IDS systems.

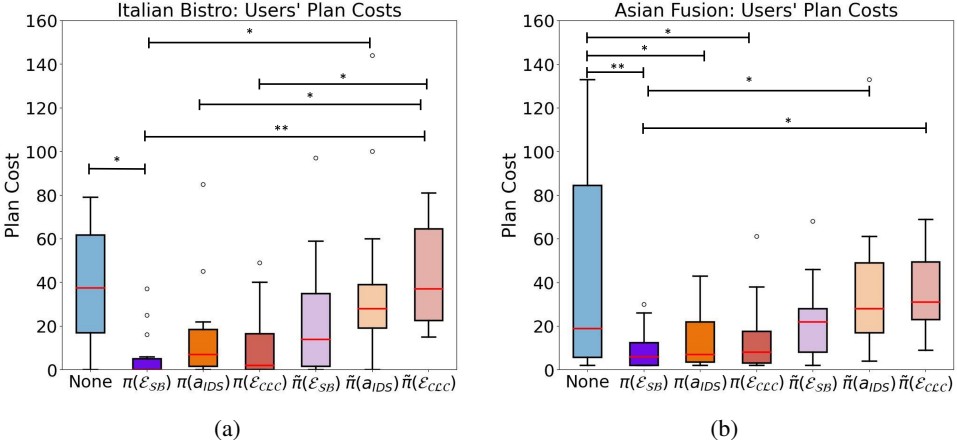

(a)

(b)

Figure 3: User Plan Cost across all conditions for the two themed games within the *IDS Stage* of the user study. Statistical significance is reported as: *p <0.05, **p <0.01, ***p <0.001 for all figures.

user performance. Overall, we observe that in both games participants in the $\pi(\mathcal{E}_{\mathcal{SB}})$ conditions, subgoal-based explanations from an optimal IDS system, have the best overall task performance in comparison to the other conditions. Additionally, out of the explanation-based conditions, we observe $\mathcal{E}_{\mathcal{CLC}}$ explanations to lead to the highest plan cost under suboptimal IDS across both games. Specifically, participants in $\pi(\mathcal{E}_{\mathcal{SB}})$ had significantly lower *UPC* compared to participants in the None condition for both the Italian Bistro game (t(95)=-3.46, $p < 0.05$) and the Asian Fusion game (t(101)=-4.00, $p < 0.01$). We additionally observe that participants in the $\pi(\mathcal{E}_{\mathcal{SB}})$ condition had significantly lower *UPC* than participants in the $\tilde{\pi}(a_{IDS})$ condition for the Asian Fusion game (t(101)=8.17, $p < 0.05$). Moreover, we observe that participants in $\pi(\mathcal{E}_{\mathcal{SB}})$ had significantly lower *UPC* compared to participants in the $\tilde{\pi}(\mathcal{E}_{\mathcal{CLC}})$ in both the Italian Bistro (t(95)=4.08, $p < 0.01$) and Asian Fusion game (t(101)=3.10, $p < 0.05$). **These results support H3** by indicating that supplementing IDS outputs with

explanations grounded in subgoal information help participants understand the underlying motivation for a suggestion and therefore perform the task significantly better those who only receive $a_{IDS}$ or even causal-link based explanations $\mathcal{E}_{\mathcal{CLC}}$).

**H2:** In Figure 4, we present user plan cost (*UPC*) from the Assessment stage across each study condition. None of the participants had access to IDS recommendations in this game, and the results allow us to assess how prior exposure to $\mathcal{E}_{\mathcal{SB}}$ impacts user performance once IDS recommendations are unavailable. Overall, we observe that both $\pi(\mathcal{E}_{\mathcal{SB}})$ and $\tilde{\pi}(\mathcal{E}_{\mathcal{SB}})$ conditions have the lowest *UPC* compared to the other study conditions, including prior work's $\mathcal{E}_{\mathcal{CLC}}$. In fact, participants in the $\pi(\mathcal{E}_{\mathcal{SB}})$ condition, those previously received $\mathcal{E}_{\mathcal{SB}}$ explanations under an optimal IDS system, have significantly lower *UPC* than participants in the $\tilde{\pi}(a_{IDS})$ condition (t(67)=2.83, $p < 0.05$). Similarly, we observe participants in the $\tilde{\pi}(\mathcal{E}_{\mathcal{SB}})$ condition, those who previously received $\mathcal{E}_{\mathcal{SB}}$ explanations under a suboptimal IDS

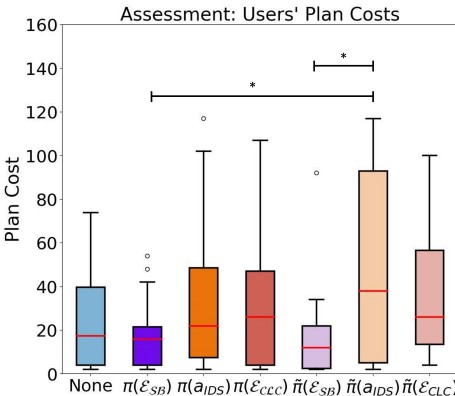

Figure 4: User Plan Costs for the assessment game in which all participants did not receive any guidance from an IDS system.

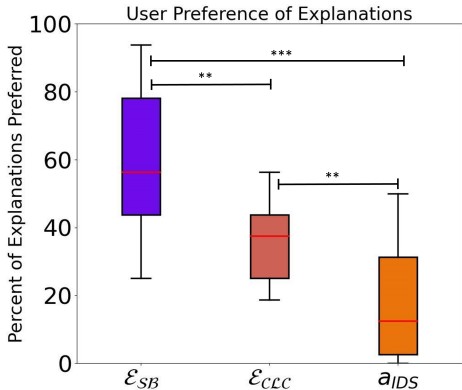

Figure 5: User perceived preferences towards both $a_{IDS}$ and $\mathcal{E}_{SB}$ in understanding the chef's next action.

system, also have significantly lower *UPC* than those who received $\tilde{\pi}(a_{IDS})$ (t(67)=2.87, $p < 0.05$). **These results support H2**, demonstrating the important role of subgoal-based explanations, $\mathcal{E}_{SB}$, in training users to understand the underlying task, compared to action-based recommendations $a_{IDS}$, even if IDS was suboptimal during training.

**H4:** To evaluate H4 we conducted an additional experiment in which we presented users with three IDS system output options, $a_{IDS}$, $\mathcal{E}_{CLC}$, and $\mathcal{E}_{SB}$, and asked them to select their preferred IDS system output[2]. Specifically, each participant was presented with 25 randomly shuffled, pre-recorded videos of optimal actions the chef would perform while preparing meals in the restaurant game. Each video was 10-15 seconds in duration and included two actions that the chef performed towards the goal. Participants were tasked with watching each video, to gain contextual understanding of which portion of the task the chef was working on, and evaluate which form of the provided IDS output, $\mathcal{E}_{SB}$, $\mathcal{E}_{CLC}$ or $a_{IDS}$, they preferred in understanding the chef's next action. Figure 5 presents the Perceived Preference (*Pref%*) metric results for the above experiment, which was analyzed with a one-way ANOVA with post-hoc Tukey Test. We observe that participants significantly prefer $\mathcal{E}_{SB}$ explanations compared to both $\mathcal{E}_{CLC}$ explanations (t(54)=-3.50, $p < 0.01$), and $a_{IDS}$ (t(54)=-6.66, $p < 0.001$). Additionally, we observe that $\mathcal{E}_{CLC}$ explanations are significantly preferred compared to $a_{IDS}$ (t(54)=-3.17, $p < 0.01$). **These results support H4** demonstrating that $\mathcal{E}_{SB}$ explanations are more frequently preferred by users, a factor that may aid in adoption of subgoal-based explanations for IDS systems.

## Discussion

Our user study findings support H1-H4, demonstrating that subgoal-based explanations ($\mathcal{E}_{SB}$) improve user task perfor-

mance, improve user ability to distinguish optimal and sub-optimal IDS recommendations, are preferred by users, and enable more robust user performance in case of IDS failure. We find the results from the Assessment stage of the study (H2) particularly surprising. All users received detailed task instructions and performed the same tutorials; the Assessment game was the 5th in the series of games, meaning that participants were reasonably familiar with the task at this stage. Yet $\mathcal{E}_{SB}$ significantly impacted performance such that in both the optimal and suboptimal IDS condition, users learned the task objectives better than users who received causal-link explanations, $\mathcal{E}_{CLC}$, and those who received only action recommendations $a_{IDS}$. These results point to important benefits explanation-based IDS systems can have in real-world deployments, highlighting that even suboptimal IDS systems can serve as a useful training tool for users when $\mathcal{E}_{SB}$ are added to IDS output. To our knowledge, this is the first use of $\mathcal{E}_{SB}$ in suboptimal IDS systems.

## Limitations and Future Work

Our work has several limitations that present opportunities for future work. First, we conducted our study only with novice users[3]. Further studies should explore whether the observed benefits of $\mathcal{E}_{SB}$ hold for expert users. Second, we conducted our study over a limited period of time, and thus factors such as long-term learning effects, fatigue, and automation bias were not fully explored. Further work is needed to fully explore the effect that explanations have on long-term IDS deployment. Third, our subgoals were predefined. Coupling our approach with autonomously identified subgoals may yield new insights. Finally, further investigation is needed to see the benefits of $\mathcal{E}_{SB}$ when subgoals are more hierarchical. For example, in complex tasks there may exist multiple hierarchical goals, or actions may satisfy multiple goals simultaneously. A more extensive comparison of various types of explanations is required in this space.

---

[2]Since our previous study was between-subjects and participants were only exposed to one type of IDS, $\mathcal{E}_{SB}$, $\mathcal{E}_{CLC}$, or $a_{IDS}$, we conducted an additional within-subjects study with 20 participants from AMT (Male=11, Female=9, mean=36.4 SD=8.7) to measure user preference between the three types of IDS system outputs.

---

[3]Our domain differs from both real world cooking and the Overcooked online game, and thus has significant novelty to all users.

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
