# OpenReview forum: "Subgoal-Based Explanations for Unreliable Intelligent Decision Support Systems"
_icaps-conference.org/ICAPS/2022/Workshop/XAIP — XAIP 2022_

### Official Review · Reviewer_ktDV · 2022-04-21
**review of "Subgoal-Based Explanations for Unreliable Intelligent Decision Support Systems"**

**Rating:** 8
**Confidence:** 4

**Review:**

The paper suggests explaining planning decisions by indicating the sub-goal that the action aims to satisfy.
The approach has been implemented, and evaluated in a user-study, comparing both objective performance and participant preferences against baselines (prior approach and no explanation). The evaluation also explores settings where the plan is optimal/sub-optimal.
Results show some benefits of the proposed approach.

One limitation of the current approach (which is acknowledged in the paper) is that the sub-goals are predefined in the planning domain. An interesting avenue for future work would be to explore sub-goals at different granularity levels, and perhaps allow the user to explore the hierarchy interactively.

The paper is well-written, and the work seems both novel and fairly mature. I think this work will make for an interesting discussion in the workshop and would be happy to see it presented.

---

### Official Review · Reviewer_SXs8 · 2022-04-28
**A novel type of explanations based on sub-goals for sub-optimal IDS**

**Rating:** 7
**Confidence:** 3

**Review:**

The authors propose a new explanation type called subgoal-based explanations in the setting of sub-optimal and non-robust Intelligent Decision Support Systems (IDS). The aim of the explanation type is to serve as a training benefit for users to (i) determine when to trust that a recommended action is optimal, and (ii) make better decisions in the absence of a recommended action through an enhanced understanding of the task due to prior provided explanations. The proposed explanation contributes to improved user task performance and is preferred over other explanation types by users in the study performed.

The proposed explanation type is intuitively simple and straightforward. The objective of the explanation is to guide naive users rather than the domain experts that most of the current explanation types cater to. The proposed method does not assume that the IDS system is ideal or optimal. The approach is measured in four different dimensions: (i) Are users able to reject more suboptimal action recommendations? (ii) Are users able to make better decisions in case the IDS breaks down and becomes unavailable? (iii) Do users perform the task better with subgoal explanations? (lower plan cost) (iv) Do users prefer the explanations over other explanation types?

There are, however, a few points I'd like to highlight:

1. It would be helpful to the readers if the authors discuss how do contrastive explanations relate to the proposed explanations. Would it help to have contrastive explanations as a baseline for comparison? It would also help to clarify why CLC explanations are most relevant.

2. Is rejecting a suboptimal action always a good decision for the users? It is possible that the user ends up making a decision that is worse than the suboptimal recommended action. It'll be nice to see a study of how many optimal and suboptimal decisions (different from the ones that the IDS suggests) a user makes after it rejects the suboptimal action recommended by the IDS with subgoal explanations.

3. In the restaurant game planner description, I suggest adding the reasons for why a horizon of 35 was prefixed.

4. From the explanations, it seems that replanning is needed to generate explanations if the user decides to reject the recommended action. Is that correct or does it replan after every action performed by the user? Should replanning cost (maybe in terms of time) be added to the overall cost for the task?

5. A discussion on how rejected or accepted recommended actions by the user relate to the trust of the user on the IDS would be interesting.

6. Did the participants involved in the study know the probability of the IDS recommendations being accurate? It would be helpful to conduct studies with IDS systems with different accuracies (currently only done with 85%).

9. In future work, it'll be interesting to see how subgoal explanations will perform in domains with dead-ends or reversible actions. Consider a situation where the user rejects the optimal action suggested by IDS assuming it to be suboptimal and performs a worse sub-optimal action. This worse sub-optimal action may either be impossible to recover from or may require reversible actions or re-performing the actions in the correct order. Can subgoal explanations also output a confidence level that will help a user to identify such critical actions and trust the IDS recommended action more?




There are minor grammatical mistakes that need to be corrected for easy readability and to avoid confusion:

1. The notation for p-value in empirical evaluation coincides with the notation for probability p with which a recommended action is changed to a suboptimal action.

2. In figure 2, the meaning of asterisks needs to be added to the caption.

3. In the abstract, "a suboptimal actions" --> "suboptimal actions"

4. In the introduction, in the 3rd paragraph and 2nd line: "characters" --> "characteristics"?

5. In the introduction, in the 4th paragraph and 3rd line: the task performance is not negatively impacted by the sudden absence of the previously available recommendations -->  the task performance is not negatively impacted by the sudden absence of the recommendations. "previously available" should be removed as the previous recommendations are available but it is the current recommendations that become unavailable.

6. In the paragraph above related work: "broadly applicable across" --> "are broadly applicable across"

7. In figure 1 caption: the statement that the planner will replan for a new plan gives the wrong notion that it replans as soon as the user rejects the recommended action. However, the replanning occurs after the user has performed an action (even though different from the one recommended by IDS). The statement can be modified to "the planner will replan for a new plan for subsequent action suggestions..".

8. In the planning problem definition: the notation for model M is different for the transition function (mathcal notation is not used).

9. In the paragraph below Hypothesis 1: "Specifically, that with the aid" --> "Specifically, we hypothesize that with the aid"

10. In the paragraph above Restaurant Game planner, in the last second line: Given a recommendation, the user can "choose either" to conform...

11. Figure 2 caption has errors: User optimal action conformance and "suboptimal" action avoidance percentages for participants that received the "three types of explanations" from suboptimal IDS systems.

12. I suggest changing the word "condition" to "study condition" throughout the paper.

---

### Meta-Review · Program_Chairs · 2022-04-30

**Recommendation:** Accept
**Confidence:** 4

**Metareview:**

The paper proposes a novel approach for generating explanations for sub-optimal IDS systems.

Both reviewers agree that that the methods described are technically sound and the paper is in a fairly mature stage. It would be a valuable addition to the workshop. As you move forward, we suggest you take into account the reviewers’ comments, especially those of reviewer 1 as they highlight some interesting points.

Thank you for submitting to the workshop. We are looking forward to your presentation.

---

### Decision · Program_Chairs · 2022-04-30

Accept